# Altered corticospinal excitability of scapular muscles in individuals with shoulder impingement syndrome

**Ya-Chu Chung[1], Chao-Ying Chen[2], Chia-Ming Chang[3], Yin-Liang Lin[1], Kwong-Kum Liao[4], Hsiu-Chen Lin[3], Wen-Yin Chen[1], Yea-Ru Yang[1], Yi-Fen Shih[1]** *

**1** Department of Physical Therapy and Assistive Technology, National Yang Ming Chiao Tung University, Taipei, Taiwan, **2** School of Physical Therapy, Chang Gung University, Tao-Yuan City, Taiwan, **3** Department of Physical Therapy, China Medical University, Taichung, Taiwan, **4** Neurological Institute, Taipei Veterans General Hospital, Taipei, Taiwan

* yfshih@nycu.edu.tw

**Data Availability Statement:** All relevant data are within the paper and its Supporting Information files.

## Abstract

The purpose of this study is to assess and compare corticospinal excitability in the upper and lower trapezius and serratus anterior muscles in participants with and without shoulder impingement syndrome (SIS). Fourteen participants with SIS, and 14 without SIS were recruited through convenient sampling in this study. Transcranial magnetic stimulation assessment of the scapular muscles was performed while the participants were holding their arm at 90 degrees scaption. The motor-evoked potential (MEP), active motor threshold (AMT), latency of MEP, cortical silent period (CSP), activated area and center of gravity (COG) of cortical mapping were compared between groups using the Mann-Whitney U tests. The SIS group demonstrated following significances, higher AMTs of the lower trapezius (SIS: 0.60 ± 0.06; Comparison: 0.54 ± 0.07, $p = 0.028$) and the serratus anterior (SIS: 0.59 ± 0.04; Comparison: 0.54 ± 0.06, $p = 0.022$), longer CSP of the lower trapezius (SIS: 62.23 ± 22.87 ms; Comparison: 45.22 ± 14.64 ms, $p = 0.019$), and posteriorly shifted COG in the upper trapezius (SIS: 1.88 ± 1.06; Comparison: 2.76 ± 1.55, $p = 0.048$) and the serratus anterior (SIS: 2.13 ± 1.02; Comparison: 3.12 ± 1.88, $p = 0.043$), than the control group. In conclusion, participants with SIS demonstrated different organization of the corticospinal system, including decreased excitability, increased inhibition, and shift in motor representation of the scapular muscles.

## 1 Introduction

Shoulder impingement syndrome (SIS) is the most common shoulder disorder accounting for at least 40% of all shoulder problems [1]. Multiple factors contribute to SIS and one of them is scapular dyskinesis [2, 3]. Scapular dyskinesis indicates that the position and movements of the scapula on the thoracic cage are aberrant. Typically, the scapula rotates upward and externally and tilts posteriorly during shoulder elevation. With adequate scapular movements, acromion would move away from the humeral head during shoulder movement. However, in

**Funding:** This work was partially funded by the Ministry of Science and Technology, Taiwan (MOST110-2410-H-A49A-515).

**Competing interests:** The authors have declared that no competing interests exist.

patients with SIS, the angles of scapular upward rotation and posterior tilt are insufficient, which may lead to impingement of subacromial tissues [3, 4].

Scapular movements and position are controlled by scapular muscles. Patients with SIS show different amplitude and timing of scapular muscle activation during movements. Through monitoring the electromyography (EMG) signals, previous studies found decreased muscle activation of the lower trapezius and the serratus anterior, and exaggerated muscle activation of the upper trapezius during shoulder elevation in athletes or individuals with SIS [5, 6]. Cools et al. [7] also found that athletes with SIS demonstrated delayed muscle activation of the upper and middle trapezius muscles compared to athletes without symptoms.

Different motor patterns may result from reorganization of the central nervous system. Although there is no structural damage in the central nervous system in patients with musculoskeletal injuries, chronic pain and prolonged symptoms are believed to alter the central nervous system [8]. The reorganization of the central nervous system may be also associated with the lack of treatment effectiveness and persistence of symptoms in some patients [9–11]. Using transcranial magnetic stimulation (TMS), recent studies have found that individuals with musculoskeletal pain, such as low back pain [12–14], chronic ankle instability [15], rotator cuff tendinopathy [16] and anterior cruciate ligament injury [17], demonstrated discrepancies in organization of the corticospinal systems, including changes in the location of the center of gravity of a cortical motor map, lower corticospinal excitability, and higher cortical inhibition [11–17]. Ngomo et al. [16] further demonstrated lower corticospinal excitability in the affected infraspinatus when compared with the non-painful side, and a relationship between the duration of pain and differences in corticospinal excitability in patients with unilateral rotator cuff tendinopathy. However, some of these variables were not reported in TMS studies of shoulder muscles such as differences in active motor threshold [12] and in intracortical inhibition [13], and it is unclear whether there are differences in the corticospinal system of other scapular muscles in patients with SIS. Therefore, the purpose of this study is to assess and compare the corticospinal excitability, intracortical inhibition and motor representation of the upper and lower trapezius, and the serratus anterior in participants with and without SIS.

## 2 Materials and methods

### 2.1 Setting and participants

Fourteen participants with unilateral SIS (7 males, 7 females; age: 24.64 ± 2.13 years old) and 14 sex, age, and hand-dominance matched participants without current shoulder or neck pain (7 males, 7 females; mean age: 23.64 ± 2.37 years old) were recruited from the local community through convenient sampling in this study (effect size: 0.724, alpha level: 0.05, power: 0.8, calculated based on the TMS data of the upper trapezius from our pilot study). The participants were only included if they were aged between 20–45 years. The inclusion criteria for the SIS were 1) chronic shoulder pain or tenderness on the greater tuberosity of the humerus for at least 3 months; 2) painful arc between 60∘ and 120∘ with active shoulder elevation; 3) positive in two of the following impingement tests: Neer impingement test, Hawkins-Kennedy impingement test, and empty can test, examined by a licensed physical therapist [18]. The exclusion criteria were: 1) a history of shoulder dislocation or traumatic injuries; 2) a history of shoulder surgery in the past six months; 3) a history of neurological diseases; 4) a history of seizure; 5) metal implants in head, neck, or chest and 6) other contraindications for receiving TMS assessment [19]. All participants were informed and signed a consent form before commencement of the study. Demographic data including weight, height, and the intensity of the most pain experienced during movement (using visual analogue scale) were collected before corticospinal excitability assessment. This study was approved by the Institutional Review

Board of Taipei Veterans General Hospital (2012-07-009A), in the spirit of the Helsinki Declaration. This experiment was conducted at the Musculoskeletal and Sports Sciences Laboratory, National Yang Ming Chiao Tung University, and the Neurological Institute, Taipei Veterans General Hospital, Taiwan.

## 2.2 Corticospinal excitability assessment

The TMS (MagStim 200 stimulator, MagStim Company, Wales, UK) was used to assess the corticospinal excitability in three scapular muscles. To record the muscle activity of the upper and lower trapezius and serratus anterior through surface EMG (Neuropack M1 MEB-9200, Nihon Kohden, Tokyo, Japan), two EMG electrodes were placed 2.0 cm apart and parallel to the direction of muscle fibers on each muscle belly [20]. The location of electrodes were at the medial 1/3 of the line connecting the spinous process of the 7[th] cervical vertebra and the acromial angle for the upper trapezius; and at the midpoint from the point between the spinous processes of the 7[th] and 8[th] thoracic vertebra, to the root of the scapular spine for the lower trapezius. For the serratus anterior, the participant was asked to lift their arm forward to 90°, and the electrodes were placed horizontally along the axillary line at the level of inferior angle of scapula, and in front of the latissimus dorsi [20]. The EMG signals were sampled at 1,500 Hz and high-pass filtered of 20 Hz. To determine the locations for stimulation on the scalp, participants were asked to wear a size-appropriate swimming cap on which the tester drew a grid with a scale of 1 cm. To match an individualized grid, the investigator first identified the center between the eyebrows and the inion to draw the Y-axis (anteroposterior axis) and identified external canals of the ears to draw the X-axis (mediolateral axis). The intersection of these two axes was the grid origin called Cz. Then, the gird of swimming cap was aligned with the individual grid, and a 12 cm*12 cm grid was over the tested hemisphere. The grid intersections were used to determine the location and guide the coil moves for stimulation.

Participants were asked to hold the affected arm at 90-degree scaption (arm elevation in the scapular plane) during whole TMS assessment (Fig 1). This testing position was used because we found it difficult to elicit TMS responses in the resting position, and the 90-degree scaption is a functional position for the shoulder complex at which our target muscles would generate about 10–15% muscle activation for TMS assessment [24]. The orientation of the 70-mm figure-of-eight TMS coil was 45 degrees to the previously defined Y-axis while TMS pulses are delivered. The stimulation started at 70–80% of the maximum stimulator output (MSO) at an approximate grid intersection where the underlying cortex controls the targeted muscle, and the inter-stimulation interval was set as 10 seconds. The coil would be moved around the surrounding grid intersections to determine the most excitable location, or the hot spot, where the stimulation would trigger the highest motor-evoked potential (MEP). Once the hot spot was located, the stimulation intensity was reduced gradually with a decrement of 5% MSO to find the lowest stimulator intensity which could elicit the peak-to-peak amplitudes of MEPs equal or over 100μV in three out of six trials. The above test was performed with individual muscle and the lowest intensity at hot spot, represent as %MSO, for the target muscle is called active motor threshold (AMT).

The testing parameters to investigate the corticospinal system for scapular muscles in this study including AMT, amplitude of MEP (amplitude), cortical silent period (CSP), latency of MEP (latency), activated area of cortical mapping (mapping area), and center of gravity of mapping area (COG). Higher AMT indicates that corticospinal excitability is decreased and a stronger stimulation is needed to trigger the action potential by recruiting more motor units of a target muscle. For better measurement reliability, the MEP amplitudes were calculated by averaging the five highest peak-to-peak EMG values out of 10 single-pulse TMS trials tested at 120%AMT at the hot spot in each scapular muscle (Fig 2) [21]. A greater amplitude means

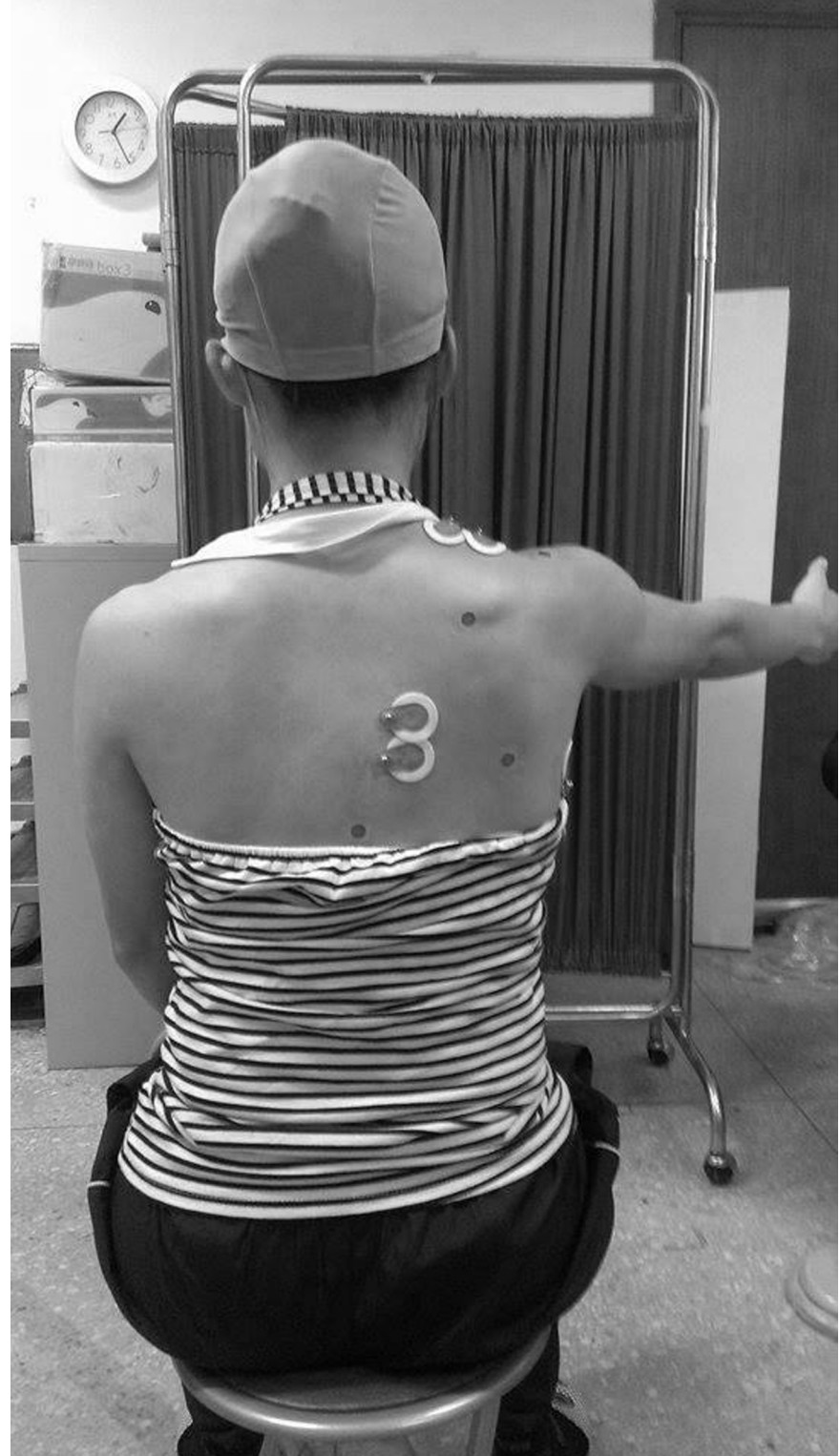

**Fig 1. TMS testing position.** The participant held the arm at 90-degree scpation elevation during TMS assessment.

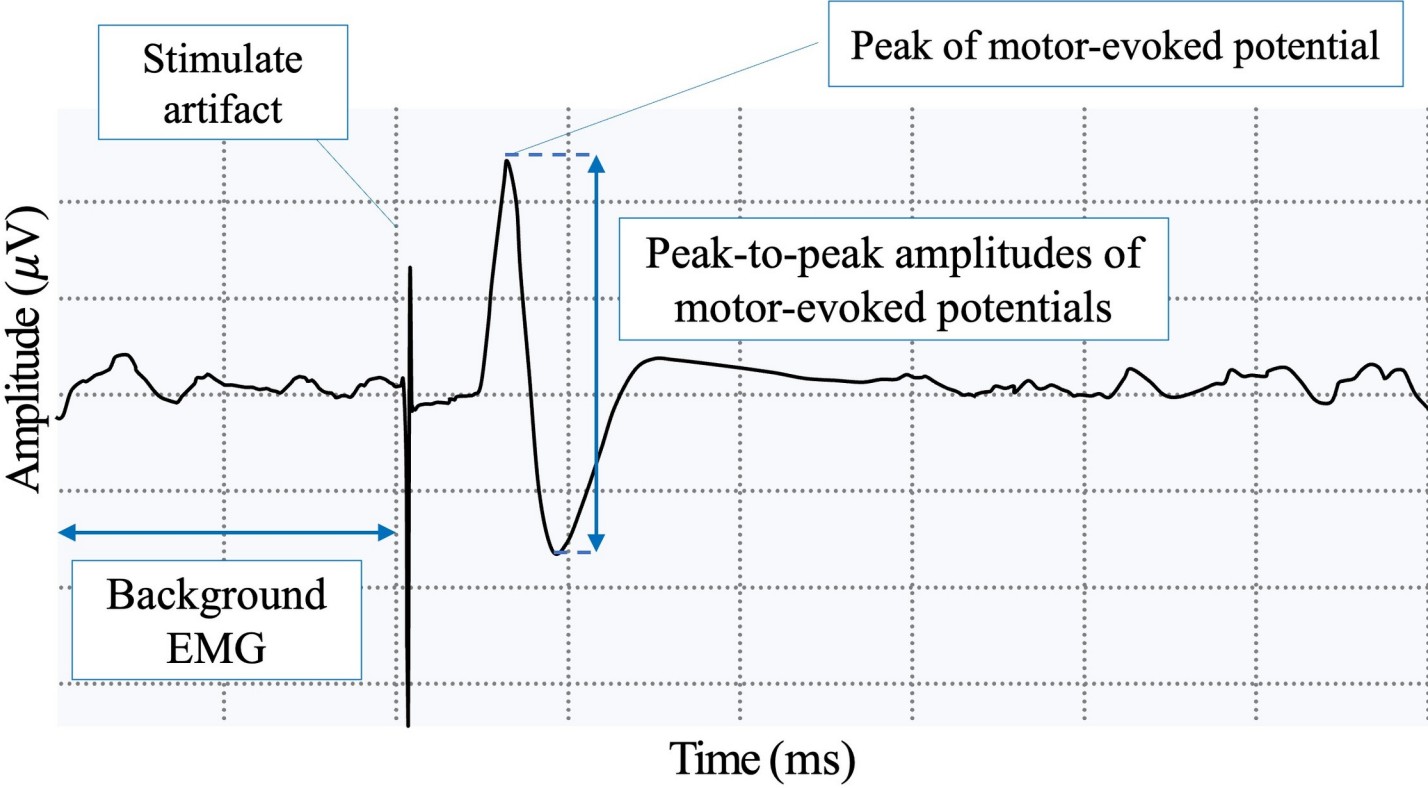

**Fig 2. A typical motor-evoked potential of upper trapezius.**

increased corticospinal excitability. The latency of MEP reflects the conductivity of the corticospinal system and was calculated by the time difference between the TMS-induced artifact and the beginning of the waveform of MEP. CSP indicates the EMG activities were suppressed temporarily after MEP when the TMS pulse is delivered during active muscle contract. The definition of intracortical CSP was defined as the time from the end of MEP to the return of EMG activity. CSP represents the inhibition mechanisms of the corticospinal system [22]. To determine the mapping area which controls each scapular muscle, TMS assessment was performed using 120% AMT. The tested spots of mapping started at the hot spot and move to the next spot in the grid along the X-axis/ Y axis. The mapping area was the range that the continuous tested spots which showed latency, MEP over 11μV and CSP in the waveform, in two out of four trials [21, 23]. Lastly, the COG of this mapping area was calculated using the following equation (Xi: location in X-axis of tested point; Yi: location in Y-axis of tested point; MEPi: corresponding amplitude of MEP; n: numbers of tested point in the mapping), and reflects a spatial average of cortical presentation of a muscle between [23]:

$$\mathrm{COG_X} = \frac{\sum_{i=1}^{n}(\mathrm{X_i} \times \mathrm{MEP_i})}{\sum_{i=1}^{n}\mathrm{MEP_i}}$$

$$\mathrm{COG_Y} = \frac{\sum_{i=1}^{n}(\mathrm{Y_i} \times \mathrm{MEP_i})}{\sum_{i=1}^{n}\mathrm{MEP_i}}$$

## 2.3 Statistical analysis

Statistical analyses were performed using SPSS 19.0 (SPSS Inc., Chicago, USA). Because we had a small sample size and our data were not normally distributed as shown by histograms, the non-parametric Mann-Whitney U tests were used to compare the differences in demographic data and all the testing parameters between the SIS and the comparison groups. A $p$ value of less than 0.05 was considered as significant.

## 3 Results

The demographics of both the SIS and the comparison groups are listed in Table 1 and there was no significant difference in the demographics between the two groups. The testing session took around 3–4 hours. None of the participants complaint of pain during testing, but most of the participants reported fatigue during and after the measurement.

All the results of the testing parameters are included in Table 2 with the original raw data in S1 Data. The AMTs of the lower trapezius ($p = 0.028$) and the serratus anterior ($p = 0.022$) were significantly higher in the SIS group. Neither the peak-to-peak amplitude of MEP nor the latency of MEP between the two groups was significantly different in all three scapular muscles. The CSP durations of the lower trapezius were significantly longer in the SIS group than in the comparison group ($p = 0.019$).

The mapping area did not show significant differences between the two groups in all three scapular muscles (Table 2). The COGs of the mapping area of the upper trapezius ($p = 0.048$) and serratus anterior ($p = 0.043$) were significantly more posterior in the SIS group than those in the comparison group ($p = 0.048$, p = 0.043, Table 2 and Fig 3). There was no significant difference in the COG of the lower trapezius mapping area.

## 4 Discussion

This study aimed to investigate whether there were differences in the corticospinal systems for scapular muscles in participants with SIS. Our results showed that the participants with SIS had lower contralateral corticospinal excitability (higher AMT) in the lower trapezius and serratus anterior, and higher contralateral intracortical inhibition (longer CSP) in the lower trapezius, and differences in contralateral cortical representation of the upper trapezius and serratus anterior. These findings reflected that SIS was associated with different organization

**Table 1. Demographic data of participants (mean ± SD).**

|  | Comparison group | SIS group | $p$-value[b] |
|---|---|---|---|
| Participant Number (Male/Female) | 14 (7/7) | 14 (7/7) | - |
| Tested Side (Right/Left) | 13/1 | 13/1 |  |
| Age (year) | 24.64 (±2.13) | 23.64 (±2.37) | 0.062 |
| Height (cm) | 166.57 (±9.30) | 170.29 (±9.82) | 0.394 |
| Weight (kg) | 60.79 (±13.54) | 63.93 (±13.62) | 0.550 |
| BMI[a] | 21.66 (±2.89) | 21.83 (±2.52) | 0.818 |
| VAS pain scale (0–10) | - | 5.57 (1.34) | - |

[a]Weight / (Height)$^2$

[b]Mann-Whitney U test

$^*p$-value < 0.05 indicates significant difference

SIS: Shoulder impingement syndrome; SD: Standard deviation; BMI: Body mass index; VAS: Visual Analogue Scale.

**Table 2. Comparisons of corticospinal excitability between the SIS and comparison groups (mean ± SD).**

| | | Comparison group (n = 14) | SIS group (n = 14) | p-value[a] |
|---|---|---|---|---|
| AMT (%MSO) | Upper Trapezius | 0.49 ± 0.06 | 0.53 ± 0.05 | 0.062 |
| | Lower Trapezius | 0.54 ± 0.07 | 0.60 ± 0.06 | 0.028* |
| | Serratus Anterior | 0.54 ± 0.06 | 0.59 ± 0.04 | 0.022* |
| MEP amplitude (mV) | Upper Trapezius | 1.20 ± 0.98 | 1.41 ± 0.64 | 0.168 |
| | Lower Trapezius | 0.82 ± 0.52 | 1.00 ± 0.44 | 0.334 |
| | Serratus Anterior | 0.50 ± 0.33 | 0.61 ± 0.38 | 0.232 |
| CSP (ms) | Upper Trapezius | 60.02 ± 23.40 | 62.25 ± 34.66 | 0.520 |
| | Lower Trapezius | 45.22 ± 14.64 | 62.23 ± 22.87 | 0.019* |
| | Serratus Anterior | 64.50 ± 21.20 | 83.94 ± 32.36 | 0.141 |
| Latency (ms) | Upper Trapezius | 8.32 ± 1.25 | 9.01 ± 1.75 | 0.395 |
| | Lower Trapezius | 11.11 ± 1.49 | 10.42 ± 1.78 | 0.280 |
| | Serratus Anterior | 12.71 ± 2.26 | 11.93 ± 3.57 | 0.783 |
| Mapping Area (cm$^2$) | Upper Trapezius | 24.95 ± 5.26 | 22.45 ± 7.91 | 0.118 |
| | Lower Trapezius | 22.83 ± 6.80 | 19.67 ± 7.19 | 0.215 |
| | Serratus Anterior | 26.93 ± 10.20 | 24.48 ± 7.10 | 0.462 |
| COG (x, y) | Upper Trapezius | (3.41 ± 0.83, 2.76 ± 1.55) | (3.23 ± 0.48, 1.88 ± 1.06) | (0.448, 0.048*) |
| | Lower Trapezius | (3.54 ± 0.75, 3.00 ± 2.02) | (3.08 ± 0.39, 1.94 ± 1.14) | (0.060, 0.066) |
| | Serratus Anterior | (3.67 ± 0.88, 3.12 ± 1.88) | (3.37 ± 0.66, 2.13 ± 1.02) | (0.270, 0.043*) |

[a]Mann-Whitney U test

*p-value < 0.05

SIS: Shoulder impingement syndrome; SD: Standard deviation; AMT: active motor threshold; MEP: Motor evoked potential; CSP: Cortical silent period; COG: Center of gravity of mapping area.

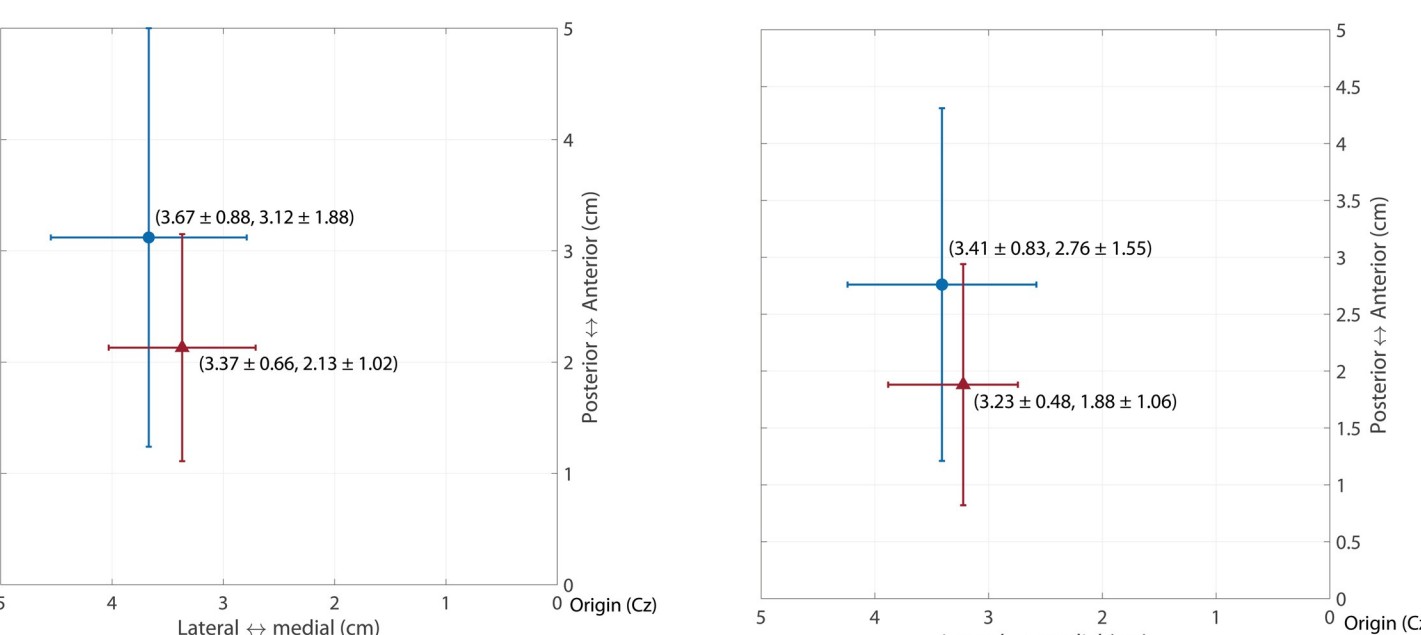

**Fig 3. The COG in the SIS and comparison groups.** Red triangle: SIS group; blue circle: Comparison group. (A) upper trapezius; (B) serratus anterior.

of the corticospinal system of the scapular muscles, which should be concerned when managing people with SIS.

The AMTs of the lower trapezius and serratus anterior were significantly higher in the SIS group than those in the comparison group, which means that a stronger stimulation is needed to trigger the action potential by recruiting more motor units of lower trapezius and serratus anterior in participants with SIS. The present finding was similar to previous studies researching the shoulder disorders. Ngomo et al. [16] found that in patients with rotator cuff tendinopathy, the AMT of the infraspinatus at the affected side was significantly higher than that at the non-affected side. Similarly, Alexander [24] demonstrated that AMT of the lower trapezius was significantly higher in patients with non-traumatic shoulder instability compared to the healthy group. One of the speculated causes of decreased corticospinal excitability is prolonged pain that changes intrinsic neuronal membrane excitability [25, 26]. The chronic pain may inhibit the motor cortex excitability that allows the spinal motor system to respond to the pain as a protective mechanism [16, 27]. It has to be noted nevertheless that chronic pain does not necessarily have the same effect as acute pain. While previous studies of systemic review and meta-analysis showed reduced corticospinal excitability in the patients with experimental acute pain [28, 29], no consistence pattern on TMS variables was observed for the chronic pain conditions [30]. In addition, our participants with SIS may change their movement patterns to avoid moving their affected shoulder due to pain. This immobilization of shoulder may also decrease the cortical excitability even within a short period of time. Previous studies identified increased resting motor threshold (RMT) following 24-hour immobilization of the elbow and fingers [31], and decreased MEP amplitudes in participants who experienced 4-day finger immobilization [32]. Both increased RMT and decreased MEP amplitudes indicated decreased corticospinal excitability.

In this study, we did not find significant differences in latency between the SIS and comparison groups. MEP latency was considered a potential indicator to represent the function of corticospinal neural transmission and to detect stressed or damaged neuronal pathway [33, 34]. In patients with musculoskeletal dysfunction, such as those with SIS, we did not expect to see impaired corticospinal tract integrity or dysfunction, which may explain why there was no difference on MEP latency between the two groups.

Measuring CSP is widely used to investigate changes of neurophysiology in patients with different diagnoses, such as Epilepsies and Parkinson's disease [35]. It can also be used to predict the motor and functional prognosis in patients with a stroke [35]. CSP is attributed by both the spinal and supraspinal inhibition mechanisms [36]. Specifically, the supraspinal mechanism contributes to the later and longer part of CSP and is medicated by gamma-aminobutyric acid (GABA) mediated inhibitory circuits [25]. In this study, the lower trapezius had longer CSP in the SIS group. This finding was similar to Bradnam et al. [26] who also found significantly longer CSP of the infraspinatus in patients with chronic shoulder pain when compared to healthy adults. This results indicated that the supraspinal inhibition may be increased due to chronic pain in participants with SIS.

We did not find differences in the mapping areas of the three scapular muscles between the SIS and comparison groups. However, the COG of the mapping area of the upper trapezius and the serratus anterior shifted posteriorly, closer to the Cz. Previous studies demonstrated different motor cortex organization represented by shifted COGs in patients with phantom limb pain after upper extremity amputation and in patients with recurrent low back pain [14, 37]. In addition, Tsao et al. found that the patients with recurrent low back pain demonstrated a more posteriorly and laterally located COG of the motor cortex as compared to the pain-free subjects, which were similar to our results [14, 38]. However, Ngomo et al. [16] did not find differences in COG locations of the infraspinatus in patients with rotator cuff tendinopathy.

Since Ngomo et al. [16] compared COGs between hemispheres within participants instead of a comparison group, it is unclear if the COG position of the hemisphere controlling the unaffected shoulder would be different while COG at the other hemisphere changed with chronic shoulder pain. This might be a potential reason that our study showed different result regarding the COG assessment.

Boroojerdi et al. [39] showed that the COG of the hand muscles matched the activated cortex area during hand clench, recorded using task functional magnetic resonance imaging [fMRI]. Therefore, locating the COG through cortical mapping is considered as a reliable way to find the corresponding motor cortex area that controls specific muscles. Alexander [24] presented the COG coordinates of the upper trapezius (3.7 ± 0.7 cm, -0.5 ± 0.9 cm), the lower trapezius (3.7 ± 0.6 cm, -0.7 ± 0.7 cm), and the serratus anterior (3.8 ± 0.5 cm, -0.6 ± 0.8 cm) based on the data of three healthy participants. Compared to the control group of this study, our result demonstrated a similar medio-lateral orientation at the X-axis in the three muscles, but a more anterior anterio-posterior position at the Y-axis. We speculated that the differences in methodologies, such as the head shapes and muscle contraction levels, and the small sample size of the previous study might contribute to the inconsistent findings of COG locations in asymptomatic participants between studies.

In the present study, we found that the three tested muscles, the upper and lower trapezius and the serratus anterior demonstrated different neurophysiological changes. These inconsistencies might be attributed to the roles and characteristics of each muscle during arm elevation activities. The lower trapezius and serratus anterior are the prime movers and stabilizers during arm elevation and majority of the studies demonstrated decreased EMG activities in these two muscles in people with shoulder impingement or pathology [5, 6, 40]. In addition to being the scapular upward rotator, the upper trapezius also contributes to clavicle movements in the beginning of shoulder elevation and was found to be over-activated by showing increased EMG activities during shoulder elevation [5, 6, 40, 41]. Therefore, most corticospinal differences were observed in the serratus anterior and/or lower trapezius, including decreases in excitability and increases in inhibition. The differences in COG, however, did not follow this pattern. The COGs located more posteriorly in the upper trapezius and serratus anterior. These two muscles demonstrated similar characteristics in firing timing. The upper trapezius fired shortly before arm movements and the serratus anterior fired in the beginning of the arm movement at 53 ms, while the lower trapezius started to activate at 349 ms after the shoulder started to move [42]. Possibly the muscles that were recruited at the beginning, but not at later stage, of movement were more likely be associated with COG changes in participants with SIS.

The present study suggested that neurophysiological changes existed in people with SIS, but there were some limitations of our methodologies. Our TMS data was recorded and analyzed without EMG activity normalization. No pre-stimulus EMG activation was recorded as the background control. This might impact interpretation of our data. We did measure EMG activation of the three target muscles (upper and lower trapezius and serratus anterior) in 12 participants (six in each group) in the preliminary study, and found no group difference in muscle activation during scaption. This however did not fully eliminate the possible effect of the group differences in EMG patterns of recruitment on our findings. Future research should consider using EMG activation or M wave to normalize the TMS data. Since the levator scapulae has similar anatomical position with the upper trapezius, there might be EMG signal crosstalk from the levator scapulae that influenced assessment results. A fine wired EMG may be used for eliminating this crosstalk in the future. During the TMS assessment, participants were asked to maintain shoulder elevation in a fixed angle and the overall assessment took 3 to 4 hours. Thus, the testing results may be influenced due to fatigue toward the end of the assessment. Furthermore, participants might experience tiredness or fail to concentrate throughout

the prolonged experimental procedure, which was also a potential factor influencing corticospinal excitability. A more efficient testing procedure or sufficient resting intervals should be considered in the future design. In addition to these limitations, all our participants experienced pain while or after performing overhead shoulder activities, but showed none or very mild symptoms during other daily tasks. It remains unclear how the patterns and frequency of pain may change corticospinal excitability and result in motor cortex reorganization. Furthermore, we did not record the history of sports for our participant. The intensity or duration of sports training may also lead to motor cortex reorganization. Since we only recruited participants aged between 20 and 45 years old, the brain plasticity at this age may impact the motor cortex organization patterns in a way that is different from other age groups after chronic pain. Finally, our results could not indicate whether the changes happen at spinal or cortical levels based on our methodologies. A study design using twin pulses of TMS might help to clarify the issue.

In conclusion, this is the first study investigating changes in the corticospinal system in scapular muscles in participants with SIS. Our results suggested the corticospinal alteration in this specific population. Future studies may need to investigate whether rehabilitation treatment could reverse these changes in the corticospinal system and whether these changes are related to the effectiveness of treatment.

## Supporting information

**S1 Data.**
(PDF)

## Author Contributions

**Conceptualization:** Kwong-Kum Liao, Wen-Yin Chen, Yi-Fen Shih.

**Data curation:** Chao-Ying Chen, Yin-Liang Lin, Wen-Yin Chen.

**Funding acquisition:** Yi-Fen Shih.

**Investigation:** Ya-Chu Chung, Kwong-Kum Liao, Hsiu-Chen Lin, Yi-Fen Shih.

**Methodology:** Kwong-Kum Liao, Yea-Ru Yang, Yi-Fen Shih.

**Resources:** Hsiu-Chen Lin, Yea-Ru Yang.

**Writing – original draft:** Ya-Chu Chung, Chao-Ying Chen, Chia-Ming Chang.

**Writing – review & editing:** Chao-Ying Chen, Chia-Ming Chang, Yin-Liang Lin, Yi-Fen Shih.

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
