## [Decision Letter · Decision Letter 0]

19 Nov 2021

PONE-D-21-33464Altered Corticospinal Excitability of Scapular Muscles in Individuals with Shoulder Impingement SyndromePLOS ONE

Dear Dr. Shih,

Thank you for submitting your manuscript to PLOS ONE. After careful consideration, we feel that it has merit but does not fully meet PLOS ONE’s publication criteria as it currently stands. Therefore, we invite you to submit a revised version of the manuscript that addresses the points raised during the review process.

We look forward to receiving your revised manuscript.

Kind regards,

Bernadette Ann Murphy, PhD

Academic Editor

PLOS ONE

Journal Requirements:

"This work was supported by the Ministry of Science and Technology, Taiwan [grant numbers MOST 106-2410-H-010-012-MY2]."

"This work was funded by the Ministry of Science and Technology, Taiwan (YF) [grant numbers MOST 106-2410-H-010-012-MY2]."

Additional Editor Comments:

As you will see, both reviewers have raised substantial concerns about the manuscript. If you believe that you can address these concerns, I would recommend revising.

Please address all concerns of both reviewers.

In particular you will need to provide additional data on the following:

1) the fact that EMG was not controlled during the scapular movements and all the differences between groups could come from the absence of EMG control. Higher EMG activity in one group could have led to in differences in AMT or CoG. You need to provide EMG activity of the three muscles during TMS testing.

2)Testing hotspot, AMT and mapping of three different muscles would certainly take 3-4 hours. Did participants report pain and fatigue during testing?

3) The location of EMG electrodes are not specified. Considering that the lower trapezius and serratus anterior are difficult to locate and test using EMG, more details are need in the methods on how the electrodes were placed and how authors did confirm positioning over these muscles. Also, a lot of cross-talk could come from adjacent muscles.

Reviewers' comments:

Reviewer's Responses to Questions

**Comments to the Author**

1. Is the manuscript technically sound, and do the data support the conclusions?

Reviewer #1: No

Reviewer #2: Partly

2. Has the statistical analysis been performed appropriately and rigorously? 

Reviewer #1: Yes

Reviewer #2: Yes

3. Have the authors made all data underlying the findings in their manuscript fully available?

Reviewer #1: No

Reviewer #2: No

4. Is the manuscript presented in an intelligible fashion and written in standard English?

Reviewer #1: Yes

Reviewer #2: Yes

5. Review Comments to the Author

Reviewer #1: PONE-D-21-33464: Altered Corticospinal Excitability of Scapular Muscles in Individuals with Shoulder Impingement Syndrome

General comments:

Yi-Fen Shih et al. compared TMS variables of three scapular muscles (serratus anterio, superior and inferior trapezius) in participants with and without impingement syndrome. They observed significant differences between multiple variables (e.g. AMT, CoGy). I have several methodological concerns that could make the results un-interpretable.

• TMS was tested during shoulder scaption and EMG was not controlled during this movement. All the differences between groups could come from the absence of EMG control. Indeed, higher EMG activity in one group could result in differences in AMT or CoG. Authors need at least to provide EMG activity of the three muscles during TMS testing.

• The TMS session was really ambitious. Testing hostpot, AMT and mapping of three different muscles would certainly take 3-4 hours. Did participants report pain and fatigue during testing?

• The location of MEG electrodes are not specified. Considering that the lower trapezius and serratus anterior are difficult to locate and test using EMG, more details are need in the methods on how the electrodes were placed and how authors did confirm positioning over these muscles. Also, a lot of cross-talks could come from adjacent muscles.

• Data are not all available in a public repository. Only means and a measure of dispersion are available in the Figures.

Specific comments:

Introduction

- Ln 40-41: I don’t think that we can assure that scapular dyskinesis is the leading factor of SIS. I would be more appropriate to indicate that this is one of the factor.

- Ln 53: Typo with reference (7)

- Ln 55: I don’t think we can use the term muscle kinetics that refer more to the biochemistry of muscle contraction rather than muscle activation. I think motor control may be a more appropriate term. Also the sentence is a bit trivial, it is clear that scapular dyskinesis suggest alteration of scapular motor patterns. Dyskinesis means modification of movement.

- Ln 63: Typo: recently should be recent

- Ln 64-66: Authors should use the term lower rather than decrease since it is cross-sectional studies. Decrease/increase infers that pain induced this effect, but we cannot conclude this from cross-sectional studies. Other studies also showed lower in intracortical inhibition level of trunk muscles in chronic low back pain compared to pain-free controls 1,2.

- Ln 77: What do authors mean by corticospinal inhibition? Do they mean intracortical inhibition? Typo: motor representation rather than presentation.

Material and methods

- Ln 78: How the effect size was calculated?

- Ln 95-98: It is important to provide more details about the exact locations of the EMG electrodes for each muscle tested. How do authors ensure that the EMG electrodes recorded muscle activity from the muscles targeted? Specifically, considering that the trapezius inferior and serratus anterior may be difficult to locate and palpate, what techniques did authors use to ensure proprer electrode placement?

- Ln 109: Why using a 90-degree scaption during TMS testing?

- Ln 117: type: grid rather than grip

- Ln 113-121:

o Did authors tested motor threshold before finding the hotspot? The order in which this is presented let is supposed this. If so, this is problematic.

o It seems like hostpsot and AMT of each muscle was measured independanlty. This takes usually a lot of time. Can authors provide how long it took to measure hotspot and AMT of each individual muscle?

- Ln 127-129: Why did only the 5 fist MEP amplitude were averaged? This is unconventional.Ln 135

- Please use references when potential mechanisms for TMS variables are stated (e.g. 3).

- Ln 139-141: Why a 11 µV MEP amplitude is used as cut-off? Why not using the same cut-off as for the AMT?

- The total duration of the session should be mentioned. This is a really long and ambitous session with three muscles tested for hotspot, AMT and mapping. How did the participants cope with such a long session? Did they report pain or fatigue? If so, how fatigue and pain may affect the results?

Results:

- It is critical to control the level of EMG that was present during TMS. Differences between groups may be driven by an increase EMG activity during the scaption movement. For exmaple, for mapping, a previous study showed that CoG was different at rest vs. in activity4. Was there differences in EMG activity for the three muscles during TMS?

 

- 1 Massé-Alarie, H., Beaulieu, L. D., Preuss, R. & Schneider, C. Corticomotor control of lumbar multifidus muscles is impaired in chronic low back pain: concurrent evidence from ultrasound imaging and double-pulse transcranial magnetic stimulation. Exp Brain Res 234, 1033-1045, doi:10.1007/s00221-015-4528-x (2016).

- 2 Massé-Alarie, H., Flamand, V. H., Moffet, H. & Schneider, C. Corticomotor control of deep abdominal muscles in chronic low back pain and anticipatory postural adjustments. Exp Brain Res 218, 99-109, doi:10.1007/s00221-012-3008-9 (2012).

- 3 Ziemann, U. et al. TMS and drugs revisited 2014. Clin Neurophysiol, doi:10.1016/j.clinph.2014.08.028 (2014).

- 4 Masse-Alarie, H., Bergin, M. J. G., Schneider, C., Schabrun, S. & Hodges, P. W. "Discrete peaks" of excitability and map overlap reveal task-specific organization of primary motor cortex for control of human forearm muscles. Hum Brain Mapp 38, 6118-6132, doi:10.1002/hbm.23816 (2017).

-

Reviewer #2: In current manuscript authors have assess and compare the corticospinal excitability, corticospinal inhibition and motor presentation of the upper and lower trapezius, and the serratus anterior in participants with and without SIS.In general manuscript is written well but for better presentation of the manuscript i would like authors to comment so following points

Abstract:

Page 3, Ln 23:Mention sampling technique used for recruitment of participants

Page 3, Ln 28: A sentence on how the data was analyzed will be useful

Page 3, Ln 29-33: Provide p-values for the results

Page 3, Ln 35: Shift “in” motor presentation

Introduction:

Page 5, Ln 64: The term “individuals with musculoskeletal pain” is very broad. Please be more specific about the population in which the changes were reported. Were these studies done on people with shoulder pain or any musculoskeletal pain at any site?

Page 5, Ln 64: Unclear what “changes in the center of gravity of a cortical map” means.

Methods:

Page 6, Ln 77: Please provide details on participant recruitment i.e., how and from where were the participants recruited for the study?

Page 6, Ln 78: Please provide more details about the inclusion criteria of participants. What was the age group that was targeted? The criteria about age group appears for the first time in the second last paragraph of the discussion section. This is needs to be mentioned in the methods section. Was pain intensity, pain duration, cause of impingement: traumatic or non-traumatic and bilateral or unilateral involvement used to screen eligible participants?

Page 6,Ln 80-83:Please provide references for the cluster of tests used for the inclusion of patients.

Page 6, Ln 82: Add “examined” before “by a licensed…” and replace “physical therapy” with “physical therapist”.

Page 8, Ln 108, 109: Was the arm held in scaption decided based on side affected or contralateral to testing hemisphere?

Page 10, Ln 150: Please provide the justification for using a non-parametric Mann-Whitney U test. Please provide information on whether the data was normally distributed.

Results:

Page 11 Ln 155: Please change the tense to past tense in Ln 155

Table 1 shows that participants pain intensity was collected using VAS scale, but this has not been mentioned in the methods section. Please provide more details about participant characteristics such as side of affected shoulder, pain duration (acute or chronic condition), does the VAS pain rating represent pain at rest or pain during movement, pain on the day of testing or average pain in past week/month.

Discussion:

Page 12 Line 170-172: Recheck sentence structure and rephrase. Include which side corticospinal excitability is increased or decreased i.e., ipsi- or contralateral. Please include the clinical implication of the findings in the first paragraph of the discussion section.

Page 15, 234: Replace “studies” with “study”

Page 17, Line 254-269: Please include some suggestions on how the limitation listed for the current study can be improved in future research.

In table two on line 387 please mention the test for the P valuse as done in line 380 for table 1.

Please improve the figure 2 ,3A and 3B quality as currently they arr blurry.

6. PLOS authors have the option to publish the peer review history of their article (what does this mean?). If published, this will include your full peer review and any attached files.

Reviewer #1: **Yes: **Hugo Massé-Alarie

Reviewer #2: No

---

## [Author Response · Author response to Decision Letter 0]

13 Jan 2022

Reviewer #1

General comments:

1. Yi-Fen Shih et al. compared TMS variables of three scapular muscles (serratus anterio, superior and inferior trapezius) in participants with and without impingement syndrome. They observed significant differences between multiple variables (e.g. AMT, CoGy). I have several methodological concerns that could make the results un-interpretable. TMS was tested during shoulder scaption and EMG was not controlled during this movement. All the differences between groups could come from the absence of EMG control. Indeed, higher EMG activity in one group could result in differences in AMT or CoG. Authors need at least to provide EMG activity of the three muscles during TMS testing.

Ans: Thank you for pointing out this issue. Because this investigation was conducted between 2014 and 2015 when using EMG activity to normalize the TMS data was less popular, we did not collect the EMG data when measuring TMS. We added discussion regarding the lack of EMG control in the limitations: “Our TMS data was recorded and analyzed without EMG activity control. This might impact some of the TMS variables such as AMT and CoG.” (P16 Ln271-274)

2. The TMS session was really ambitious. Testing hostpot, AMT and mapping of three different muscles would certainly take 3-4 hours. Did participants report pain and fatigue during testing?

Ans: Thank you for raising this question. The whole testing session took around 3-4 hours and most of the participants reported not much pain, but fatigue. This information is added in the Results (P10, Line 169-170) and discussed in the limitations. (P17 Ln277-283)

3. The location of MEG electrodes are not specified. Considering that the lower trapezius and serratus anterior are difficult to locate and test using EMG, more details are need in the methods on how the electrodes were placed and how authors did confirm positioning over these muscles. Also, a lot of cross-talks could come from adjacent muscles.

Ans: Thank you for reviewer’s suggestion. The details of the EMG electrode locations are added in the 2.2 Corticospinal excitability assessment, Procedure (P6, Ln101-107).

4. Data are not all available in a public repository. Only means and a measure of dispersion are available in the Figures.

Ans: Raw data are now available in supplement 1.

Specific comments:

Introduction

5. Ln 40-41: I don’t think that we can assure that scapular dyskinesis is the leading factor of SIS. I would be more appropriate to indicate that this is one of the factor.

Ans: Thank you for reviewer’s suggestion. The sentence is revised accordingly:” Multiple factors contribute to SIS and one of them is scapular dyskinesis (P3 Ln40).”

6. Ln 53: Typo with reference (7)

Ans: Thank you for reviewer’s reminder. This typo has been corrected. (P5 Ln 51)

7. Ln 55: I don’t think we can use the term muscle kinetics that refer more to the biochemistry of muscle contraction rather than muscle activation. I think motor control may be a more appropriate term. Also the sentence is a bit trivial, it is clear that scapular dyskinesis suggest alteration of scapular motor patterns. Dyskinesis means modification of movement.

Ans: Thank you for reviewer’s comment. We have deleted the redundant sentence.

8. Ln 63: Typo: recently should be recent

Ans: Thank you for reviewer’s reminder. This typo is corrected. (P4 Ln59)

9. Ln 64-66: Authors should use the term lower rather than decrease since it is cross-sectional studies. Decrease/increase infers that pain induced this effect, but we cannot conclude this from cross-sectional studies. Other studies also showed lower in intracortical inhibition level of trunk muscles in chronic low back pain compared to pain-free controls 12.

Ans: Thank you for reviewer’s valuable suggestion. The description is revised accordingly, using “lower” and “higher” instead of decreases or increases. (P4 Ln63-64)

10. Ln 70: What do authors mean by corticospinal inhibition? Do they mean intracortical inhibition? Typo: motor representation rather than presentation.

Ans: Thank you for reviewer’s reminder. We changed the “corticospinal inhibition” to “intracortical inhibition”, and the typo is corrected. (P4 Ln69-70)

Material and methods

11. Ln 78: How the effect size was calculated?

Ans: This effect size was calculated based on the AMT of the upper trapezius from our pilot study. This information is added. (P5 Ln 78)

12. Ln 95-98: It is important to provide more details about the exact locations of the EMG electrodes for each muscle tested. How do authors ensure that the EMG electrodes recorded muscle activity from the muscles targeted? Specifically, considering that the trapezius inferior and serratus anterior may be difficult to locate and palpate, what techniques did authors use to ensure proprer electrode placement?

Ans: Thank you for reviewer’s suggestion. The details of the EMG electrode locations are added in the 2.2 Corticospinal excitability assessment, Procedure (P6, Ln101-107).

13. Ln 109: Why using a 90-degree scaption during TMS testing?

Ans: Thank you for the question. We found it difficult to elicit TMS responses in the resting position, and the 90-degree scaption is a functional position for the shoulder complex at which our target muscles would generate about 10-15% muscle activation for TMS assessment. (P8, Ln119-122)

14. Ln 117: type: grid rather than grip

Ans: Thank you. This typo is corrected. (P9, Ln126)

15. Ln 113-121: Did authors tested motor threshold before finding the hotspot? The order in which this is presented let is supposed this. If so, this is problematic.

Ans: Thank you for raising this issue. We started with 70-80% of the maximum stimulator output (MSO) to find the hot spot first, and then gradually decreased the intensity to find the lowest stimulator intensity as the active motor threshold (AMT). The sentence is corrected to clarify the confusion. (P9, Ln126-131)

16. Ln 113-121: It seems like hostpsot and AMT of each muscle was measured independanlty. This takes usually a lot of time. Can authors provide how long it took to measure hotspot and AMT of each individual muscle?

Ans: Thank you for raising this concern. We did measure the hotspot and AMT independently for each muscle. Because the hotspots and AMTs of these three muscle are not far from each other, it was not too troublesome to do it one by one. It generally took us about an hour to do all three. 

17. Ln 127-129: Why did only the 5 fist MEP amplitude were averaged? This is unconventional.

Ans: Thank you for the question. We average the five highest EMP out of 10 trials, not the first five. The sentence is corrected. (P9, Ln139-140)

18. Ln 135: Please use references when potential mechanisms for TMS variables are stated (e.g. 3).

Ans: Thank you for the suggestion. The reference is added. (P9 Ln148)

19. Ln 139-141: Why a 11 µV MEP amplitude is used as cut-off? Why not using the same cut-off as for the AMT?

Ans: Thank you for raising the question. This criterion came from Wassermann et al. (1992)’s protocol, and the reference is added. (P9 Ln 152)

20. The total duration of the session should be mentioned. This is a really long and ambitous session with three muscles tested for hotspot, AMT and mapping. How did the participants cope with such a long session? Did they report pain or fatigue? If so, how fatigue and pain may affect the results?

Ans: Thank you for the suggestion. The whole testing session took around 3-4 hours and most of the participants reported not much pain, but fatigue. This information is added in the Results (P10, Line 168--170) and discussed in the limitations. (P17 Ln277-283)

Results:

21. It is critical to control the level of EMG that was present during TMS. Differences between groups may be driven by an increase EMG activity during the scaption movement. For exmaple, for mapping, a previous study showed that CoG was different at rest vs. in activity. Was there differences in EMG activity for the three muscles during TMS?

Ans: Thank you for raising the concern. We realized the importance of EMG activity control for the TMS data. Because this investigation was conducted between 2014 and 2015 when using EMG activity to normalize the TMS data was less popular, we did not collect the EMG data when measuring TMS. We added discussion regarding the lack of EMG control in the limitations: “Our TMS data was recorded and analyzed without EMG activity control. This might impact some of the TMS variables such as AMT and CoG.” (P18 Ln322-324)

Reviewer #2: 

In current manuscript authors have assess and compare the corticospinal excitability, corticospinal inhibition and motor presentation of the upper and lower trapezius, and the serratus anterior in participants with and without SIS.In general manuscript is written well but for better presentation of the manuscript i would like authors to comment so following points

Abstract:

1. Page 3, Ln 23:Mention sampling technique used for recruitment of participants

Ans: Thank you for reviewer’s comment. We used convenient sampling (leaflets) to recruit participants. The information is added. (Abstract, Ln24)

2. Page 3, Ln 28: A sentence on how the data was analyzed will be useful

Ans: Thank you for reviewer’s suggestion. We used Mann-Whitney U tests to analyze the data. The information is added. (Abstract, Ln28)

3. Page 3, Ln 29-33: Provide p-values for the results

Ans: Thank you for reviewer’s suggestion. All p-values are added. (Abstract, Ln30-34)

4. Page 3, Ln 35: Shift “in” motor presentation

Ans: Thank you for reviewer’s reminder. The “in” is added. (Abstract, Ln35)

Introduction:

5. Page 5, Ln 64: The term “individuals with musculoskeletal pain” is very broad. Please be more specific about the population in which the changes were reported. Were these studies done on people with shoulder pain or any musculoskeletal pain at any site?

Ans: Thank you for reviewer’s comment. These included low back pain, chronic ankle instability, rotator cuff tendinopathy, and anterior cruciate ligament injury. The information is added. (P4, Ln60-61)

6. Page 5, Ln 64: Unclear what “changes in the center of gravity of a cortical map” means..

Ans: Thank you for reviewer’s reminder. We rephrased the sentence to clarify the confusion as” changes in the location of center of gravity…” (P4, Ln62)

Methods:

7. Page 6, Ln 77: Please provide details on participant recruitment i.e., how and from where were the participants recruited for the study?

Ans: Thank you for reviewer’s reminder. We used convenient sampling (leaflets) to recruit participants from the local community. The information is added. (P5, Ln77)

8. Page 6, Ln 78: Please provide more details about the inclusion criteria of participants. What was the age group that was targeted? The criteria about age group appears for the first time in the second last paragraph of the discussion section. This is needs to be mentioned in the methods section. Was pain intensity, pain duration, cause of impingement: traumatic or non-traumatic and bilateral or unilateral involvement used to screen eligible participants?

Ans: Thank you for reviewer’s suggestion. We add detailed inclusion criteria in the “participant” section. (P5 Ln79-84) 

Page 6,Ln 80-83:Please provide references for the cluster of tests used for the inclusion of patients.

Ans: Thank you for the suggestion. The reference 16 is added.

Magee DJ, Manske RC. Orthopedic Physical Assessment: Elsevier - Health Sciences Division; 2021.

9. Page 6, Ln 82: Add “examined” before “by a licensed…” and replace “physical therapy” with “physical therapist”.

Ans: Thank you for reviewer’s suggestion. The sentence is revised accordingly. (P5 Ln83-84)

10. Page 8, Ln 108, 109: Was the arm held in scaption decided based on side affected or contralateral to testing hemisphere?

Ans: Thank you for raising the concern. The testing arm was the affected side. The sentence is revised to clarify the confusion. (P7 Ln117)

11. Page 10, Ln 150: Please provide the justification for using a non-parametric Mann-Whitney U test. Please provide information on whether the data was normally distributed.

Ans: Thank you for raising the concern. A non-parametric test was used because of the small sample size and some our data were not normally distributed as shown by histograms. This information is added. (P10, Ln162-163)

Results:

12. Page 11 Ln 155: Please change the tense to past tense in Ln 155

Ans: Thank you. The typo is corrected accordingly. (P10, Ln168)

13. Table 1 shows that participants pain intensity was collected using VAS scale, but this has not been mentioned in the methods section. Please provide more details about participant characteristics such as side of affected shoulder, pain duration (acute or chronic condition), does the VAS pain rating represent pain at rest or pain during movement, pain on the day of testing or average pain in past week/month.

Ans: Thank you for the suggestion. The information is added in the Methods (chronic pain for at least 3 months, the most pain experienced during movement etc.) (P5, Ln80-90) and Table 1. 

Discussion:

14. Page 12 Line 170-172: Recheck sentence structure and rephrase. Include which side corticospinal excitability is increased or decreased i.e., ipsi- or contralateral. Please include the clinical implication of the findings in the first paragraph of the discussion section.

Ans: Thank you for reviewer’s suggestion. We revised the paragraph as suggested “Our results showed that the participants with SIS had lower contralateral corticospinal excitability (higher AMT) in the lower trapezius and serratus anterior, and higher contralateral intracortical inhibition (longer CSP) in the lower trapezius, and changes in contralateral cortical representation of the upper trapezius and serratus anterior. These findings reflected that SIS was associated with reorganization of corticospinal system of the scapular muscles, which should be concerned when managing people with SIS.” (P12 Ln185-190)

15. Page 15, 234: Replace “studies” with “study”

Ans: Thank you. The “studies” is replaced by the “study”. (P16 Ln270)

16. Page 17, Line 254-269: Please include some suggestions on how the limitation listed for the current study can be improved in future research.

Ans: Thank you for reviewer’s suggestion. We add some information regarding how to improve the study methods in the “limitation”. (highlighted, P16 Ln 270-293)

17. In table two on line 387 please mention the test for the P valuse as done in line 380 for table 1.

Ans: Thank you for the suggestion. The information is added in Table 2.

18. Please improve the figure 2 ,3A and 3B quality as currently they arr blurry.

Ans: Thank you for reviewer’s suggestion. We re-do the figures with better resolutions. (Figure 2 and 3)

---

## [Decision Letter · Decision Letter 1]

4 Feb 2022

PONE-D-21-33464R1Altered Corticospinal Excitability of Scapular Muscles in Individuals with Shoulder Impingement SyndromePLOS ONE

Dear Dr. Shih,

Thank you for submitting your manuscript to PLOS ONE. After careful consideration, we feel that it has merit but does not fully meet PLOS ONE’s publication criteria as it currently stands. Therefore, we invite you to submit a revised version of the manuscript that addresses the points raised during the review process. Please submit your revised manuscript by Mar 21 2022 11:59PM. If you will need more time than this to complete your revisions, please reply to this message or contact the journal office at plosone@plos.org. Please include the following items when submitting your revised manuscript:A rebuttal letter that responds to each point raised by the academic editor and reviewer(s). You should upload this letter as a separate file labeled 'Response to Reviewers'.A marked-up copy of your manuscript that highlights changes made to the original version. You should upload this as a separate file labeled 'Revised Manuscript with Track Changes'.An unmarked version of your revised paper without tracked changes. You should upload this as a separate file labeled 'Manuscript'.

We look forward to receiving your revised manuscript.

Kind regards,

Bernadette Ann Murphy, PhD

Academic Editor

PLOS ONE

Additional Editor Comments:

Please ensure that you address Reviewer 1's comments in this revision.

Reviewers' comments:

Reviewer's Responses to Questions

**Comments to the Author**

1. If the authors have adequately addressed your comments raised in a previous round of review and you feel that this manuscript is now acceptable for publication, you may indicate that here to bypass the “Comments to the Author” section, enter your conflict of interest statement in the “Confidential to Editor” section, and submit your "Accept" recommendation.

Reviewer #1: (No Response)

Reviewer #2: All comments have been addressed

2. Is the manuscript technically sound, and do the data support the conclusions?

Reviewer #1: No

Reviewer #2: Yes

3. Has the statistical analysis been performed appropriately and rigorously? 

Reviewer #1: Yes

Reviewer #2: Yes

4. Have the authors made all data underlying the findings in their manuscript fully available?

Reviewer #1: (No Response)

Reviewer #2: Yes

5. Is the manuscript presented in an intelligible fashion and written in standard English?

Reviewer #1: Yes

Reviewer #2: Yes

6. Review Comments to the Author

Reviewer #1: PONE-D-21-33464: Altered Corticospinal Excitability of Scapular Muscles in Individuals with Shoulder Impingement Syndrome

General comments:

I thank authors for considering my comments. However, there are some authors’ responses for which I am not convinced and other elements I would like them to address.

- About EMG control, in 2014-2015, it was well known that EMG background amplitude (representing the net motoneuronal excitability) was critical to control in TMS research considering the MEP amplitude represent the excitability at both cortical and spinal levels. Authors may control for EMG by measuring pre-stimulus EMG background to determine if it was similar between groups for each muscle tested. For example, if participants of the SIS group recruit more the lower trapezius during the task, it may explain the longer duration of the CSP or the difference in AMT. Further analyses are required to ensure that the results are not due to different patterns of recruitment in the different groups.

- I encourage authors to review the literature and report TMS studies comparing patients with and without musculoskeletal pain for cortical and corticospinal control. For example, differences in AMT 1 and in intracortical inhibition 2. Considering that these variables were not reported in TMS studies of shoulder muscles, it may help for the discussion of the results.

- About the use of the 5 highest MEP out of 10, why did the authors used these methods? Could they use references and discuss how it could impact on the results?

- Ln. 64-67: In this sentence, it should be clear that the difference was with the non-painful side, and not with a control group.

- About sample size calculation: what was the alpha and beta levels used?

- About the position of the electrodes, authors should provide a reference for these electrode sites.

- About the justification of the scaption position, please refer to a study for the EMG level stated (10-15% MVC).

- Ln 202-203: Please refer to systematic reviews and meta-analysis (e.g. 3,4) and it is important to be cautious to assume that chronic pain has the same effect as acute pain. A systematic review of chronic pain did not find any consistent pattern on TMS variables 5 conversely to experimental acute pain which reduces corticospinal excitability 3,4.

- Ln 231-232: Authors can discuss/described a bit more the findings of Tsao et al. that showed a difference in CoG position in the M1 map in low back pain compared to a painfree group 6,7.

 

-

- 1 Strutton, P. H., Theodorou, S., Catley, M., McGregor, A. H. & Davey, N. J. Corticospinal excitability in patients with chronic low back pain. J Spinal Disord Tech 18, 420-424, doi:00024720-200510000-00008 [pii] (2005).

- 2 Massé-Alarie, H., Beaulieu, L. D., Preuss, R. & Schneider, C. Corticomotor control of lumbar multifidus muscles is impaired in chronic low back pain: concurrent evidence from ultrasound imaging and double-pulse transcranial magnetic stimulation. Exp Brain Res 234, 1033-1045, doi:10.1007/s00221-015-4528-x (2016).

- 3 Burns, E., Chipchase, L. S. & Schabrun, S. M. Primary sensory and motor cortex function in response to acute muscle pain: A systematic review and meta-analysis. Eur J Pain, doi:10.1002/ejp.859 (2016).

- 4 Rohel, A. et al. The effect of experimental pain on the excitability of the corticospinal tract in humans: A systematic review and meta-analysis. Eur J Pain 25, 1209-1226, doi:10.1002/ejp.1746 (2021).

- 5 Parker, R. S., Lewis, G. N., Rice, D. A. & McNair, P. J. Is Motor Cortical Excitability Altered in People with Chronic Pain? A Systematic Review and Meta-Analysis. Brain Stimul 9, 488-500, doi:10.1016/j.brs.2016.03.020 (2016).

- 6 Tsao, H., Danneels, L. A. & Hodges, P. W. ISSLS prize winner: Smudging the motor brain in young adults with recurrent low back pain. Spine (Phila Pa 1976) 36, 1721-1727, doi:10.1097/BRS.0b013e31821c4267 (2011).

- 7 Tsao, H., Galea, M. P. & Hodges, P. W. Reorganization of the motor cortex is associated with postural control deficits in recurrent low back pain. Brain 131, 2161-2171, doi:10.1093/brain/awn154 (2008).

-

Reviewer #2: Thanks for addressing my comments.Authors have satisfactorily replied to my concern and I am happy to endorse the paper.I only have a minor aesthetic comment about the quality of figure 3 A & B .It would be nice to improve the quality for overall readership of the article.

7. PLOS authors have the option to publish the peer review history of their article (what does this mean?). If published, this will include your full peer review and any attached files.

Reviewer #1: **Yes: **Hugo Massé-Alarie

Reviewer #2: No

---

## [Author Response · Author response to Decision Letter 1]

21 Feb 2022

Reviewer #1

General comments:

1. I thank authors for considering my comments. However, there are some authors’ responses for which I am not convinced and other elements I would like them to address. About EMG control, in 2014-2015, it was well known that EMG background amplitude (representing the net motoneuronal excitability) was critical to control in TMS research considering the MEP amplitude represent the excitability at both cortical and spinal levels. Authors may control for EMG by measuring pre-stimulus EMG background to determine if it was similar between groups for each muscle tested. For example, if participants of the SIS group recruit more the lower trapezius during the task, it may explain the longer duration of the CSP or the difference in AMT. Further analyses are required to ensure that the results are not due to different patterns of recruitment in the different groups.

Ans: Thank you for reviewer’s suggestion. Unfortunately the raw EMG set has been deleted, we were unable to retrieve it for further analysis. Some of the participants (6 in each group) were tested in the preliminary study for EMG activation of the three target muscles during scaption, and we found no group difference in EMG activities of the upper and lower trapezius and serratus anterior (p values ranging from 0.26 to 0.99). This information was added in the final paragraph of the Discussion. (Line 278-284)

2. I encourage authors to review the literature and report TMS studies comparing patients with and without musculoskeletal pain for cortical and corticospinal control. For example, differences in AMT 1 and in intracortical inhibition 2. Considering that these variables were not reported in TMS studies of shoulder muscles, it may help for the discussion of the results.

Ans: Thank you for reviewer’s valuable suggestion. References are added as suggested. (P4 Ln60-69)

3. About the use of the 5 highest MEP out of 10, why did the authors used these methods? Could they use references and discuss how it could impact on the results?

Ans: Thank you for pointing out the question. This method was chosen for better measurement reliability based on the Cavaleri et al.’s review, and this information is added. (P8 Ln142-145) 

4. Ln. 64-67: In this sentence, it should be clear that the difference was with the non-painful side, and not with a control group.

Ans: Thank you for reviewer’s suggested. The sentence is revised accordingly to avoid the confusion. (P4 Ln65)

5. About sample size calculation: what was the alpha and beta levels used?

Ans: Thank you for reviewer’s reminder. The information is added (effect size: 0.724, alpha level: 0.05, power: 0.8). (P5 Ln79-80)

6. About the position of the electrodes, authors should provide a reference for these electrode sites.

Ans: Thank you for reviewer’s reminder. The reference of placing these EMG electrodes was added. (P6 Ln104 & P7 Ln111)

7. About the justification of the scaption position, please refer to a study for the EMG level stated (10-15% MVC).

Ans: Thanks for raising the question. This information is added. (P7 Ln124-125, reference 24)

8. Ln 202-203: Please refer to systematic reviews and meta-analysis (e.g. 3,4) and it is important to be cautious to assume that chronic pain has the same effect as acute pain. A systematic review of chronic pain did not find any consistent pattern on TMS variables 5 conversely to experimental acute pain which reduces corticospinal excitability 3,4.

Ans: Thank you for the valuable suggestion. This information is added. (P13 Ln205-209)

9. Ln 231-232: Authors can discuss/described a bit more the findings of Tsao et al. that showed a difference in CoG position in the M1 map in low back pain compared to a painfree group 6,7.

Ans: Thank you for reviewer’s suggestion. We add this information in the discussion. (P15 Ln239-241) 

- 1 Strutton, P. H., Theodorou, S., Catley, M., McGregor, A. H. & Davey, N. J. Corticospinal excitability in patients with chronic low back pain. J Spinal Disord Tech 18, 420-424, doi:00024720-200510000-00008 [pii] (2005).

- 2 Massé-Alarie, H., Beaulieu, L. D., Preuss, R. & Schneider, C. Corticomotor control of lumbar multifidus muscles is impaired in chronic low back pain: concurrent evidence from ultrasound imaging and double-pulse transcranial magnetic stimulation. Exp Brain Res 234, 1033-1045, doi:10.1007/s00221-015-4528-x (2016).

- 3 Burns, E., Chipchase, L. S. & Schabrun, S. M. Primary sensory and motor cortex function in response to acute muscle pain: A systematic review and meta-analysis. Eur J Pain, doi:10.1002/ejp.859 (2016).

- 4 Rohel, A. et al. The effect of experimental pain on the excitability of the corticospinal tract in humans: A systematic review and meta-analysis. Eur J Pain 25, 1209-1226, doi:10.1002/ejp.1746 (2021).

- 5 Parker, R. S., Lewis, G. N., Rice, D. A. & McNair, P. J. Is Motor Cortical Excitability Altered in People with Chronic Pain? A Systematic Review and Meta-Analysis. Brain Stimul 9, 488-500, doi:10.1016/j.brs.2016.03.020 (2016).

- 6 Tsao, H., Danneels, L. A. & Hodges, P. W. ISSLS prize winner: Smudging the motor brain in young adults with recurrent low back pain. Spine (Phila Pa 1976) 36, 1721-1727, doi:10.1097/BRS.0b013e31821c4267 (2011).

- 7 Tsao, H., Galea, M. P. & Hodges, P. W. Reorganization of the motor cortex is associated with postural control deficits in recurrent low back pain. Brain 131, 2161-2171, doi:10.1093/brain/awn154 (2008).

Reviewer #2

1. Thanks for addressing my comments. Authors have satisfactorily replied to my concern and I am happy to endorse the paper. I only have a minor aesthetic comment about the quality of figure 3 A & B. It would be nice to improve the quality for overall readership of the article.

Ans: Thank you for reviewer’s valuable comment. We further improved the quality of fig 3A & B as suggested.

---

## [Decision Letter · Decision Letter 2]

23 Mar 2022

PONE-D-21-33464R2Altered Corticospinal Excitability of Scapular Muscles in Individuals with Shoulder Impingement SyndromePLOS ONE

Dear Dr. Shih,

Thank you for submitting your manuscript to PLOS ONE. After careful consideration, we feel that it has merit but does not fully meet PLOS ONE’s publication criteria as it currently stands. Therefore, we invite you to submit a revised version of the manuscript that addresses the points raised during the review process.

Please address the few remaining issues raised by reviewer 1.

We look forward to receiving your revised manuscript.

Kind regards,

Bernadette Ann Murphy, PhD

Academic Editor

PLOS ONE

Journal Requirements:

Additional Editor Comments:

You are nearly there, just a few additional comments to address.

Reviewers' comments:

Reviewer's Responses to Questions

**Comments to the Author**

1. If the authors have adequately addressed your comments raised in a previous round of review and you feel that this manuscript is now acceptable for publication, you may indicate that here to bypass the “Comments to the Author” section, enter your conflict of interest statement in the “Confidential to Editor” section, and submit your "Accept" recommendation.

Reviewer #1: All comments have been addressed

2. Is the manuscript technically sound, and do the data support the conclusions?

Reviewer #1: Yes

3. Has the statistical analysis been performed appropriately and rigorously? 

Reviewer #1: Yes

4. Have the authors made all data underlying the findings in their manuscript fully available?

Reviewer #1: Yes

5. Is the manuscript presented in an intelligible fashion and written in standard English?

Reviewer #1: Yes

6. Review Comments to the Author

Reviewer #1: General comments:

I thank the authors for considering my comments. I would urge authors to be cautious in the language used for data interpretation. Differences between groups coming from cross-sectional studies cannot be interpreted as “changes”, “reorganisation” or “modification”. Only a longitudinal study allows to conclude this. Authors should change these terms used throughout the manuscript. Also, I think the English should be revised before publication of the manuscript.

Specific comments:

Ln 32 : Considering this is a cross-sectional study, it is not possible to conclude in a “reorganisation” of the corticospinal system since it is not known if this is a predisposing factors or a modification which occurs because of pain. Authors should use a the term “different organisation” or “suggest a reorganisation”

Ln 279-280: I would be careful in the language used to interpret the findings. The sentence “The present study proved that neurophysiological changes existed in people with SIS, but there were some limitations of our methodologies.” should be modified to “ The present study suggest […]”.

Ln 320: “suggest” instead of “supported”

7. PLOS authors have the option to publish the peer review history of their article (what does this mean?). If published, this will include your full peer review and any attached files.

Reviewer #1: **Yes: **Hugo Massé-Alarie

---

## [Author Response · Author response to Decision Letter 2]

28 Mar 2022

Reviewer #1: General comments:

I thank the authors for considering my comments. I would urge authors to be cautious in the language used for data interpretation. Differences between groups coming from cross-sectional studies cannot be interpreted as “changes”, “reorganisation” or “modification”. Only a longitudinal study allows to conclude this. Authors should change these terms used throughout the manuscript. Also, I think the English should be revised before publication of the manuscript.

Answer: Thanks for the suggestions. The manuscript has been proof read by an English native speaker, and terms such as reorganization, modification, changes have been revised according to your suggestion throughout the manuscript (highlighted).

Specific comments:

Ln 32 : Considering this is a cross-sectional study, it is not possible to conclude in a “reorganisation” of the corticospinal system since it is not known if this is a predisposing factors or a modification which occurs because of pain. Authors should use a the term “different organisation” or “suggest a reorganisation”

Answer: Thanks for the suggestion. The sentence is revised as “participants with SIS demonstrated different reorganization of the corticospinal system…”. (Ln34-35)

Ln 279-280: I would be careful in the language used to interpret the findings. The sentence “The present study proved that neurophysiological changes existed in people with SIS, but there were some limitations of our methodologies.” should be modified to “ The present study suggest […]”.

Answer: Thanks for the suggestion. The sentence is revised as “ The present study suggested […]”. (Ln 280)

Ln 320: “suggest” instead of “supported”

Answer: Thanks for the suggestion. The sentence is revised accordingly. (Ln 310)

---

## [Decision Letter · Decision Letter 3]

3 May 2022

Altered Corticospinal Excitability of Scapular Muscles in Individuals with Shoulder Impingement Syndrome

PONE-D-21-33464R3

Dear Dr. Shih,

We’re pleased to inform you that your manuscript has been judged scientifically suitable for publication and will be formally accepted for publication once it meets all outstanding technical requirements.

Kind regards,

François Tremblay, PhD

Academic Editor

PLOS ONE

Additional Editor Comments (optional):

Reviewers' comments:

Reviewer's Responses to Questions

**Comments to the Author**

1. If the authors have adequately addressed your comments raised in a previous round of review and you feel that this manuscript is now acceptable for publication, you may indicate that here to bypass the “Comments to the Author” section, enter your conflict of interest statement in the “Confidential to Editor” section, and submit your "Accept" recommendation.

Reviewer #1: All comments have been addressed

Reviewer #3: All comments have been addressed

2. Is the manuscript technically sound, and do the data support the conclusions?

Reviewer #1: Yes

Reviewer #3: Yes

3. Has the statistical analysis been performed appropriately and rigorously? 

Reviewer #1: Yes

Reviewer #3: Yes

4. Have the authors made all data underlying the findings in their manuscript fully available?

Reviewer #1: Yes

Reviewer #3: Yes

5. Is the manuscript presented in an intelligible fashion and written in standard English?

Reviewer #1: Yes

Reviewer #3: Yes

6. Review Comments to the Author

Reviewer #1: The authors addressed all my comments in the revised version of the manuscript.

I would like to thank the authors for adressing my comments.

Reviewer #3: The authors have addressed all queries raised by the reviewers. The authors have performed multiple revisions and I think the work is important. These are difficult muscles to target via TMS so the work will be of interest to groups trying to attempt this type of stimulation for shoulder muscles.

7. PLOS authors have the option to publish the peer review history of their article (what does this mean?). If published, this will include your full peer review and any attached files.

Reviewer #1: **Yes: **Hugo Massé-Alarie

Reviewer #3: No

---

## [Editor Report · Acceptance letter]

6 May 2022

PONE-D-21-33464R3 

Altered Corticospinal Excitability of Scapular Muscles in Individuals with Shoulder Impingement Syndrome 

Dear Dr. Shih:

I'm pleased to inform you that your manuscript has been deemed suitable for publication in PLOS ONE. Congratulations! Your manuscript is now with our production department. 

Kind regards, 

on behalf of

Dr. François Tremblay 

Academic Editor

PLOS ONE